# Animal abuse by falsification–Recognition amongst the veterinary profession in The Netherlands

Ineke R. van Herwijnen ◐*, Daphne G. L. van Helvoort◐, Nadieh Reinders‡, Claudia M. Vinke‡

Division of Animals in Science and Society, Faculty of Veterinary Medicine, Department of Population Health Sciences, Utrecht University, Utrecht, The Netherlands

◐ These authors contributed equally to this work.
‡ These authors also contributed equally to this work.
* i.r.vanherwijnen@uu.nl

## Abstract

Presently, Animal Abuse by (condition/ illness) Falsification (AAF), has received little scientific attention. Feigning illness in *children* for attention purposes has been studied, indicating that the involved children can suffer serious consequences. Although to date little is known about AAF, perplexing presentation has been mentioned. We aim to add to scientific information on veterinary awareness of AAF. Our exploratory survey-based research addressed veterinarians and vet technicians/ assistants, asking questions on their awareness of AAF and recognition of possible AAF signs. We found that only 12% of our 88 participants, who mainly treated companion animals, had previously received education on AAF. Despite this, 83% reported familiarity with the phenomenon of AAF. Half of the participants (51%), indicated to *likely* see AAF cases in their veterinary clinic, 5% indicated to see them with *certainty* (unsure: 32%; not to see such cases: 12%). Most often dogs and cats were indicated as a proxy (other animals: rabbits, rodent, horse). When asked how likely a participant would regard a sign/ symptom indicative of AAF, participants scored higher likeliness for those signs/ symptoms regarding client behaviours than animal/ medical aspects. Reporting of AAF in this sample was low: 92% indicated to have never reported AAF as animal abuse. Barriers to reporting AAF included a lack of knowledge on AAF and how to identify clients suffering from the condition. Our study engaged a limited number of participants in only one country, but indicates that knowledge on AAF may facilitate the recognition of this form of animal abuse.

## Introduction

Abuse by (Paediatric Condition) Falsification is related to serious and life-long outcomes [1]. It regards the infliction of harm to a child as to feign its illness and thus get

**Data availability statement:** All relevant data are within the paper and its Supporting Information files.

**Funding:** The author(s) received no specific funding for this work.

**Competing interests:** The authors have declared that no competing interests exist.

attention and/ or sympathy from people, including medical personnel [2,3]. Outcomes may include physical and mental trauma, e.g., undergoing unnecessary and often repeated surgeries, suffering malnutrition and developing distrust of important others [3–5].

The condition of people feigning illness for attention purposes, is known under different names, such as fabricated or induced illness disorder, factitious disorder, fictitious illness, illness fabrication, illness falsification and Munchausen [3,6–8]. People who suffer from the condition seemingly need attention from people (medical personnel). To get this attention they may either be dishonest about illness symptoms ('fabricated'), create these symptoms ('induced'), or both [4]. Among the suggested causes of the condition arising in a person, are those of early life experiences or a lack thereof, such as of repeated medical treatments during childhood or inadequate care provision [9]. The condition may present itself as harming oneself, or it can be 'imposed on another', a 'proxy', such as a child [1,10].

Like children, animals are expected to be a possible proxy. Animal Abuse by (Condition) Falsification (AAF) has been indicated in animals under human care [2,11–13]. Alternatively, at the least, the perplexing presentation has been indicated [7,14]. Perplexing presentation regards clinically noted signs that can be explained by multiple causes, including AAF as a causation. Perplexing presentation may be a starting point to assess if AAF underlies a case. Alternatively, it may be the only label that can validly be given, as the assessment of Abuse by (Condition) Falsification is both complex and challenging [7,14]. The complex and challenging assessment of this form of abuse, likely affects studying its incidence or prevalence. A USA-based study indicated that the true incidence or prevalence of AAF is unknown [2]. The few available scientific studies on the topic, often present cases and do not provide epidemiological data [2]. An example is a UK-based study including six cases identified by veterinarians and three cases included by the researchers as possible AAF cases [13]. In this total of nine cases, signs were noted in the animals, that corresponded with signs in child abuse cases. The respective signs regarded the animal owner's attention seeking behaviour, recovery of the animal upon owner-separation, serial incidents, real and likely factitious clinical signs, abnormal biochemical profiles, deliberately inflicted injury, tempering with surgical sites and changing one's veterinary clinic [13]. Thus, a similarity may exist in signs of AAF and Abuse by (Paediatric Condition) Falsification.

To date few scientific studies are available on AAF. Interestingly, one study made a distinction between feigning illness for attention purposes and malingering [12]. Generally, malingering differs from illness falsification, by the perpetrator having instrumental purposes, such as feigning illness to avoid work, or to obtain medications of abuse [15]. Obtainment of medications of abuse in the veterinary clinical setting has been studied, resulting in the reporting on five veterinary cases of suspected or confirmed malingering with this purpose [12]. This instrumental purpose is obviously also of high importance to the veterinary profession, but out of scope of our study.

In our study we aim to add to scientific information on veterinary awareness of AAF, following the conclusion from Oxley & Feldman [2], that there is a need for studying the phenomenon. AAF can cause prolonged suffering in animals, their

deaths and serial cases have been described, indicating that a perpetrator can cause serious harm to multiple animals [11,13]. Hence, our aim is to create insight into awareness of the condition amongst the Dutch veterinary profession. Theoretical relevance is in identifying a scientific questionnaire approach that can aid in assessing awareness levels in a profession that has a high responsibility regarding the recognition of signs forewarning AAF. Practical relevance is in assessing which awareness is present or lacking as to improve present day education to students of veterinary science. This exploratory research addresses the awareness of AAF in the veterinary profession in The Netherlands and the extent to which signs and symptoms are recognised as likely indicators of AAF.

## Methods

### Ethical considerations

We approached Dutch veterinary professionals via email and internet platforms where they professionally reside, asking them to participate of their own free will, after reading information on the study and providing informed consent in the survey module by clicking a survey button that indicated 'I consent', before continuing to the survey. The topic of this study regards an emotionally burdening topic, which we specifically addressed in the participant information. No minors were included in the study. We sought and received ethical approval from the Science-Geosciences Ethics Review Board of Utrecht University for involved human participants on August 18th, 2025 (ERB Review Science-25–0080). The survey was accessible from September 22th, 2025 to October 18th, 2025.

### Participants recruitment and survey

We recruited Dutch veterinary professionals, that is veterinarians and vet technicians/ assistants via email, and internet platforms where they professionally reside. A Faculty of Veterinary Medicine digital newsletter going out to 4,900 veterinary professionals, a Facebook group for veterinarians with 2,000 members, a Facebook group for vet technicians/ assistants with 2,900 members and a Facebook group for veterinary professionals with 4,700 members published a short notice asking participants to fill out the study survey.

Participants were asked to fill out a survey with general questions on AAF to assess awareness and experience with the phenomenon, assess reporting of AAF and barriers to reporting. We presented our participants with the barriers to reporting, as listed by Williams et al. [16] and endorsed by at least a third of their study sample of N = 215 veterinarians completing survey items on *general* animal abuse. Amongst their survey items were items on barriers to the reporting of animal abuse cases. The barrier items received 1,024 item endorsements and were presented in their article as a percentage of the 215 participants, followed by the number of item endorsements. We present our responses similarly as a percentage of participants, followed by the number of item endorsements. We adapted the original items from Williams et al. [16], by replacing 'fear' for 'concern' in four of the nine items, to increase suitability for the Dutch veterinary profession.

After the survey items on reporting of AAF, our second survey section followed. This second section was construed of a list of possible AAF signs and symptoms. For each of these signs and symptoms we asked participants to indicate the likeliness of pointing at AAF, on a five-point Likert scale ranging from highly unlikely (1) to highly likely (5). The signs and symptoms were derived from two AAF articles [2,13] and four articles from human child literature, which were selected for their specific listing of signs and symptoms of Abuse by (Paediatric Condition) Falsification [1,3,4,6]. S1 File presents the full survey and S2 Table indicates our choices for signs and symptoms and the appearance of similar signs and symptoms in the sources we used to base this list on. After presenting our participants with the single signs and symptoms, this section also questioned how participants would regard combinations of signs and symptoms.

In the third and final survey section our participants answered questions on the background of the participants and their veterinary clinics. (See S1 File for all questions and S1 Data for the data file).

## Data preparation and statistical analysis

Data from Qualtrics® was gathered in Microsoft® Excel. Statistical analysis of the data was performed with SPSS® for Windows®, version 29 (SPSS Inc, Chicago, USA).

For the two open questions 'Which signs would make you think of AAF?' and 'What would help you to report AAF?', the answers were coded and categorised manually, through thematic analysis and using description focused coding.

We assessed how likely the participants felt signs and symptoms indicative of AAF in two blocks. The first block regarded animal and medical aspects, the second block regarded client behaviours on a five-point Likert scale ranging from highly unlikely (1) via moderately unlikely (2), neutral (3), moderately likely (4) to highly likely (5). The same five-point Likert scale was then used to assess if the number of present signs and symptoms affected AAF likeliness ratings, asking 'How likely would you rate a combination of signs as indicative of AAF?' and assessing each of these three options: two signs, three to five signs, more than five signs. We present median (range) for each of the Likert scale items.

Finally, we tested with Wilcoxon tests, if a difference in likeliness scores existed between two subsamples of the participants, for the signs/ symptoms regarding animal/ medical aspects and the client behaviours. The subsamples regarded: a) those participants indicating a (certain/ likely) occurrence of AAF in their veterinary clinic and b) those participants that indicated AAF not to occur in their clinic, or being uncertain if it did. We regarded P-values <0.05 as significant.

## Results

### Participant characteristics

Our study sample of 88 participants consisted mainly of females (90.9%, N = 80; males: 9.1%, N = 8) and the majority were veterinarians (68.2%, N = 60; vet technicians/ assistants: 31.8%, N = 28). The majority worked in a veterinary clinic providing service to companion animals and their owners (88.6%, N = 78; mixed companion animals and other animals: 8.0%, N = 7; horses: 3.4%, N = 3). See S3 Table for all details on gender, ages and the professional/ clinic background.

The participants indicated mostly to have never received education on AAF (88.6%, N = 78; with 11.4%, N = 10 indicating they had followed some form of education on the topic).

### AAF awareness in the veterinary profession in The Netherlands

When asking participants if they were familiar with one or both of the terms 'animal abuse by fabricated or induced illness' and 'Munchausen syndrome by Proxy' the majority answered that they were familiar with one or both terms: 89.8% (N = 79). 8.0% (N = 7) indicated they were not and 2.3% (N = 2) indicated not to know. Next, we presented a definition of AAF, namely: 'An induced/ fabricated condition in an animal which has been induced/ fabricated by a caretaker, such as the owner. This caretaker has a sickly (morbid) need for attention by making an animal ill or pretending such illness.' We asked if participants were familiar with this phenomenon. Here too, the majority answered positively: 83.0% (N = 73). 14.7% (N = 13) indicated not to be familiar with the phenomenon and 2.3% (N = 2) indicated 'Don't know'.

### Experience with suspected AAF cases in Dutch veterinary clinics

To assess participant experience with the phenomenon, we asked if they see AAF cases in their veterinary clinic. 51.2% (N = 45) indicated to likely see cases. 4.6% (N = 4) indicated to certainly see cases, 12.5% (N = 11) indicated to certainly not see cases in their clinic. A relatively high percentage of 31.8% (N = 28) indicated 'don't know'. When we asked participants to indicate a yearly average of cases that they suspected of AAF, most indicated none (84.1%, N = 74). Of those that indicated a yearly average of cases, the range of yearly cases was one to three cases. One case a year was indicated by 10.2% (N = 9), with one of these participants indicating that ten animals were involved in the one case. 2.3% (N = 2) indicated two cases per year. 3.4% (N = 3) indicated three cases a year. Participants could indicate which animals were proxy

in the suspected cases. Most mentioned dogs (N = 16) and cats (N = 12). Other mentioned animals were rabbits (N = 4), a rodent (N = 1) and a horse (N = 1).

### Recognition of AAF signs and symptoms

Thematic analysis of 173 label indications by 72 participants answering the open question 'Which signs would make you think of AAF?', rendered 38 different labels in 6 categories, listed in Table 1. See S4 Table for all entries.

The category of 'complaint characteristics' (42.8%, N = 74) and 'contact characteristics' (34.1%, N = 59) held high numbers of labels and had high label counts. For the first category the label 'vague or inexplicable complaints - difficult to check or confirm medically' had the highest percentage of occurrence (20.2%, N = 35). For the second category 'recurrent contact moments/ visits with one or multiple professionals' had the highest percentage of occurrence (18.5%, N = 32). Not all participants answered this facultative question. One participant wrote: 'None. I know that it [abuse by falsification] exists in humans, but never considered that people do this to animals. I'd rather think of abuse, but not with the objective to gain attention'.

After asking the open question on AAF recognition on which we presented the results above, we presented the two blocks of signs and symptoms indicative of AAF. We present median and ranges in Table 2. We assessed with Wilcoxon tests if there was a difference between the N = 49 participants that indicated that AAF (certainly/ likely) occurred in their clinic and the N = 39 who indicated this not to be the case or being uncertain. We found a difference for three signs/ symptoms. For each of these, the likeliness scores were higher for participants that indicated AAF occurred in their clinic: 'Incompatibilities between medical history and clinical findings', 'Difficulty of diagnosis, rare or unsuspected disease pattern', and the client behaviour: 'Relatively often visiting the clinic with this or multiple animals' (all P < 0.05 and the table indicates the Wilcoxon Z-values).

The number of present signs and symptoms affected how likely participants thought a case to regard AAF, with the median for two signs: 2 (1–4), three to five signs: 4 (1–5), more than five signs: 5 (1–5).

### Reporting of AAF signs and symptoms

With regard to reporting of AAF, we asked participants a) if they ever reported AAF as animal abuse, b) if they worked with a network partner, such as the police and c) which barriers to reporting they experienced and what would help them to report a suspected AAF case. The vast majority of the participants indicated to have never reported AAF as animal abuse (92.1%, N = 81). 4.5% (N = 4) reported AAF as animal abuse, 3.4% (N = 3) didn't know. Of the 4.5% (N = 4) participants that reported AAF as animal abuse, 3.4% (N = 3) did so in cooperation with a network partner such as the police, 1.1% (N = 1) did so independently.

We assessed barriers to reporting, following Williams et al.'s [16] list of barriers that veterinarians experienced to reporting general animal abuse. Our 88 participants scored the barriers to reporting AAF a total of 253 times, as multiple barriers could be indicated by one participant. Table 3 indicates how in our sample the highest frequencies were for 'Lack of knowledge of available resources' (56.8%, N = 50), 'Lack of accepted standards in identification' (53.4%, N = 47) and 'Concerns about breaking client confidentiality' (46.6%, N = 41). The first two items were indicated in our sample at similar percentages as in the study by Williams et al. [16]. The other barriers were indicated at relatively lower percentages in our sample.

Thematic analysis of 93 label indications by 63 participants answering the open question 'What would help you to report AAF?', rendered 22 different labels in 4 categories, listed in Table 4. See S5 Table for all entries.

### Discussion

We aimed to assess awareness on Animal Abuse by (Condition) Falsification (AAF) in the Dutch veterinary profession and found that over eighty percent indicated familiarity with the phenomenon, although under fifteen percent indicated

**Table 1. Categories and labels distilled from 72 participants regarding signs that would make one think of AAF.**

| Category | Label | % (N of N = 173) |
|---|---|---|
| Animal keeping characteristics | Multiple animals | 1.2% (2) |
| | Rehoming/ relinquishment of animals | 0.6% (1) |
| | Request for euthanasia at relatively young age | 0.6% (1) |
| | | *2.3% (4)* |
| Disease and condition characteristics | Inexplicable injuries/ uncommon fractures | 5.8% (10) |
| | Signs of diarrhoea/ GI and/ or starving – famished animal | 2.9% (5) |
| | Signs of itchiness, allergy, relating to the immune system | 1.2% (2) |
| | Signs of lameness | 1.2% (2) |
| | Signs of poisoning | 1.2% (2) |
| | Sign PU-PD (PolyUria-PolyDipsia) | 0.6% (1) |
| | | *12.7% (22)* |
| Complaint characteristics | Vague or inexplicable complaints – difficult to check or confirm medically | 20.2% (35) |
| | (Unlikely) repeatedly new or changing symptoms, multiple problems | 6.9% (12) |
| | Repetitive small complaints/ visits with a healthy animal | 6.9% (12) |
| | Clinical abnormality absence upon medical/ laboratory/ additional checks | 2.3% (4) |
| | High frequency of complaints | 2.3% (4) |
| | Complaints that disappear upon clinical admission | 1.2% (2) |
| | Mentioning symptoms in the manner that a disease is described | 1.2% (2) |
| | Owner diagnosed animal before appointment | 1.2% (2) |
| | Uncommon seriousness of complaints | 0.6% (1) |
| | | *42.8% (74)* |
| Treatment course characteristics | Incorrect administration of medication/ no improvement on medication | 2.3% (4) |
| | Slowed healing/ recovery or even worsening despite treatment | 2.3% (4) |
| | Self-medicating (doctoring) the animal | 1.2% (2) |
| | Animal is reportedly ill, when owners visit no clinical findings at first, but clinical findings present at a second visit | 0.6% (1) |
| | | *6.4% (11)* |
| Contact characteristics | Recurrent contact moments/ visits with one or multiple professionals | 18.5% (32) |
| | High concern ventilation by owner and/ or lengthy in conversations | 3.5% (6) |
| | Unwilling to take advice from a veterinarian/ no interest in solutions that will aid quick recovery | 2.9% (5) |
| | Asking for medication by phone, insisting on medication prescription, or ordering it via internet by themselves | 1.7% (3) |
| | Conversations by client on medical issues of others than animal, e.g., relatives, self, other animals than patient | 1.2% (2) |
| | Insisting, being friendly but also complaining | 1.2% (2) |
| | Requests for unnecessary (invasive) examination | 1.2% (2) |
| | Asking specifically to talk to the veterinarian | 0.6% (1) |
| | Being in close contact with personnel | 0.6% (1) |
| | Blaming the veterinarian if the animal does not recover and/ or claiming that no one can help the animal | 0.6% (1) |
| | Challenging the diagnosis | 0.6% (1) |
| | Deviant behaviour of owner(s) upon discussion of possible causes of a condition | 0.6% (1) |
| | Insisting on hospitalisation or longer hospital admission | 0.6% (1) |
| | Social media behaviour | 0.6% (1) |
| | | *34.1% (59)* |

*(Continued)*

**Table 1.** (Continued)

| Category | Label | % (N of N=173) |
|---|---|---|
| Client characteristics | Client with personality disorder (characteristics)/ deviance in mental health state | 1.2% (2) |
| | A profession in (health)care of client | 0.6% (1) |
| | | 1.7% (3) |

**Table 2.** Likeliness indication (median, range) for AAF signs and symptoms in the N=88 participants and Wilcoxon Z-values and P-values for the difference between the subsample indicating AAF (certainly/ likely) occurred in their clinic (N=49) and those indicating it did not/ were uncertain (N=39).

| Signs and symptoms regarding animal and medical aspects | Median (range) | Wilcoxon Z-value | P-value |
|---|---|---|---|
| Difficulty of diagnosis, rare or unsuspected disease pattern | 4 (1–5) | **−2.117** | **0.034** |
| Erratic or toxic drug blood levels | 4 (1–5) | −1.539 | 0.124 |
| Gastro intestinal complaints for more than two weeks without a definitive diagnosis | 4 (1–5) | −0.546 | 0.585 |
| Incompatibilities between medical history and clinical findings | 4 (1–5) | **−2.603** | **0.009** |
| Inexplicable medical symptoms | 4 (1–5) | −0.782 | 0.434 |
| Neurological complaints, e.g., epileptiform activity, lesser alertness (incl. coma) | 4 (1–5) | −1.417 | 0.157 |
| Persistent or recurrent illnesses for which a cause cannot be found | 4 (1–5) | −0.022 | 0.982 |
| Recovery of animal/ symptoms when hospitalized | 4 (1–5) | −1.614 | 0.106 |
| Recurrent illnesses in which poisoning may factor in | 4 (1–5) | −1.106 | 0.269 |
| Repeated hospitalisations and vigorous medical/ veterinary evaluations of victim without definitive diagnoses | 4 (1–5) | −0.399 | 0.690 |
| Unlikely medical history | 4 (1–5) | −0.233 | 0.816 |
| Relatively many animals deceased with client | 3.5 (1–5) | −1.315 | 0.189 |
| Inexplicable intolerance of treatment or poor response to treatment | 3 (1–5) | −1.857 | 0.063 |
| Physiologic or laboratory parameters are noticeable or not fitting with patient profile | 3 (1–5) | −0.758 | 0.449 |
| Poor recovery if animal is with client | 3 (1–5) | −0.960 | 0.337 |
| Recurrent illnesses in which nutrition, nutritional absorption, nutritional state may factor in | 2 (1–5) | −0.292 | 0.771 |
| Recurrent illnesses in which suffocation may factor in | 2 (1–5) | −0.395 | 0.693 |
| **Signs and symptoms regarding client behaviours** | | | |
| Much knowledge of the presented illness or generally of the medical/ veterinary field | 4 (2–5) | −0.101 | 0.920 |
| Relatively often talking about own illness or illness of relatives | 4 (1–5) | −0.650 | 0.516 |
| Relatively often visiting the clinic with this or multiple animals | 4 (1–5) | **−2.868** | **0.004** |
| Unexpected response in communication, such as anger upon referral | 4 (1–5) | −0.064 | 0.949 |
| Expression of concern by relatives or other professionals | 3 (1–5) | −0.489 | 0.625 |
| Little concern expressed over painful examinations/ surgery | 3 (1–5) | −1.430 | 0.153 |
| Perpetrator is known to have provided false information | 3 (1–5) | −0.167 | 0.867 |
| Relatively often talking about care burden or death of animal(s) | 3 (1–5) | −1.638 | 0.101 |
| Relatively often talking about illness in this or multiple animals | 3 (1–5) | −1.144 | 0.253 |
| Repeatedly presenting animal at various clinics | 3 (1–5) | −0.432 | 0.666 |
| Resistance of client to (possibly effective) therapy suggestions | 3 (1–5) | −1.166 | 0.244 |
| Sudden withdraw from treatment or staying away from clinic | 3 (1–5) | −1.783 | 0.075 |

**Table 3. Barriers to reporting (suspected) AAF cases as mentioned by our 88 participants and barriers to reporting _general_ animal abuse by 215 veterinarians in the study by Williams et al. [16].**

| | Regarding AAF cases in this study % (N of 253 indicated reasons) | Regarding animal abuse in Williams et al. [16] % (N of 1,024 item endorsements) |
|---|---|---|
| Lack of knowledge of available resources | 56.8% (N = 50) | 54% (N = 117) |
| Lack of accepted standards in identification | 53.4% (N = 47) | 54% (N = 116) |
| Concerns about breaking client confidentiality | 46.6% (N = 41) | 69% (N = 148) |
| A perception no action will be taken | 33.0% (N = 29) | 47% (N = 100) |
| Inexperience in dealing with misleading information provided by client | 33.0% (N = 29) | 47% (N = 102) |
| Concerns on litigation | 26.1% (N = 23) | 52% (N = 111) |
| Concern that reporting may compromise safety of victim | 18.2% (N = 16) | 62% (N = 133) |
| Concern on physical retaliation by perpetrator | 11.4% (N = 10) | 35% (N = 75) |
| Concern that client will leave the clinic | 9.1% (N = 8) | 42% (N = 90) |

**Table 4. Categories and labels distilled from 63 participants regarding what would help to report AAF.**

| Category | Description (label) | % (N of N = 93) |
|---|---|---|
| Information | More knowledge on topic and owner identification | 18.3% (17) |
| | More information | 9.7% (9) |
| | Knowing whom to contact for help/ knowing how the process of reporting works | 6.5% (6) |
| | Experience and familiarity with the condition | 2.2% (2) |
| | More knowledge on signs | 1.1% (1) |
| | Knowledge on consequences | 1.1% (1) |
| | | _38.7% (36)_ |
| Support & Assurance | Certainty of recognition/ no trust breach/ no false accusation risk | 6.5% (6) |
| | Opportunity to consult psychologist/ physician of client | 5.4% (5) |
| | Support from colleagues/ employer | 3.2% (3) |
| | Low time investment/ financial support for additional examination | 2.2% (2) |
| | Knowing that action is taken upon reporting | 2.2% (2) |
| | No need to discuss with client yourself | 1.1% (1) |
| | Assistance on recognition as to not incorrectly accuse a person | 1.1% (1) |
| | More support | 1.1% (1) |
| | | _22.6% (21)_ |
| Guidelines and protocol | (Easy access to) reporting centre/ disclosure office | 12.9% (12) |
| | Guidelines/ protocol | 10.8% (10) |
| | (Knowing) where to report | 5.4% (5) |
| | Clarity on roles in the process and consequences for the involved | 1.1% (1) |
| | | _30.1% (28)_ |
| Clarity on condition (recognition) | Flow chart for near certain diagnosis/ guide for documentation of signs and symptoms/ list of signs of the condition | 4.3% (4) |
| | Better definition | 2.2% (2) |
| | Examples of what people with the condition do to animals | 1.1% (1) |
| | A clear diagnosis (opportunity) | 1.1% (1) |
| | | _8.6% (8)_ |

The category 'Information' (38.7%, N = 36) and 'Support and assurance' (22.6%, N = 21) held high numbers of labels and had high label counts. For the first category the label of 'More knowledge on topic and owner identification' had the highest percentage of occurrence (18.3%, N = 17). For the second 'Certainty of recognition/ no trust breach/ no false accusation risk' had the highest percentage of occurrence (6.5%, N = 6). Not all participants answered this question.

previous education on AAF. Half of the participants indicated to likely see AAF cases in their veterinary clinic. Five percent indicated to see such cases with certainty, with often mentioned proxies being dogs and cats. Of the possible signs and symptoms indicative of AAF, signs and symptoms referring to client behaviours were reported spontaneously more so than signs and symptoms referring to animal/ medical aspects. The first were also rated higher likeliness of indicating AAF.

Generally, we demonstrate in this study that an awareness of the existence of AAF may be present in the Dutch veterinary profession. In the open answers regarding the participants' recognition of AAF, we may even find an indication that the difference made in child abuse cases between fabricated and induced symptoms (e.g., [4,8]), may also be recognised by the veterinary profession. Our participants reported on the one hand a possible dishonesty ('fabrication') without induction of illness and on the other hand the induction thereof. Examples of fabrication are recognisable in 'vague or inexplicable complaints – difficult to check or confirm medically', and 'repetitive small complaints/ visits with a healthy animal'. Examples of induction are recognisable in 'uncommon seriousness of complaints' and 'slowed healing/ recovery or even worsening despite treatment'.

The existence of variants of abuse by falsification is only one aspect of the complexity of the phenomenon. Also, on the perpetrator side complexity exists: the condition that leads a person to abuse a proxy by condition falsification, is recognised as a spectrum disorder [8,17]. Due to the existence of variants of the phenomenon, just like in child abuse cases [14], diagnosing AAF in animal abuse cases will be challenging. It is therefore logical that our participants indicate the need for more knowledge on AAF, and on support when dealing with suspected AAF cases. Collaboration opportunities are flagged as important, involving not only colleagues, but also professionals in the human care domain. Such a support need was also indicated in an Italian survey amongst more than 500 paediatricians when suspecting child abuse by falsification [18]. The participants to this survey indicated to more likely discuss a case with another specialist (or the parent of the victim), than to refer the case to the authorities [18]. As time may be of the essence in cases of abuse by falsification [19], further studying of how reporting can be facilitated seems of relevance to animal welfare. Possible suggestions from this study may be offering knowledge in a certain format, such as flow chart, and facilitating reporting via an (easily accessible) reporting centre/ disclosure office. Furthermore, guidelines on the assessment and management of AAF are warranted. Such guidelines could build on more general animal abuse veterinary medicine guidelines [20,21]. Creating AAF guidelines would follow their existence in human medicine, where such guidelines have been written, although not yet for all subsamples, such as adolescents [19,22].

The limitation of our study primarily regards the limited number of participants. Despite our efforts to involve a large number of veterinary professionals in The Netherlands, we managed only to include 88 participants. This may have been due to the longer length of the survey, that presented the participants with AAF signs/symptoms from human medicine, which we deemed a starting point for assessing AAF in another proxy. Another study limitation may be the unintentional involvement of higher numbers of topic-engaged participants. Consequently, the found percentages of AAF occurring likely or certainly in veterinary clinics, as per report of our participants, may *not* be regarded as prevalence numbers. Also, people inflicting AAF may or may not be seen in veterinary clinical settings. The need for attention is central to the condition that leads a person to abuse a proxy by condition falsification. Such attention may be received by visiting veterinary clinics. Yet, people may fulfil the condition-related psychological need via the internet [23,24], possibly limiting their visiting of veterinary clinics for attention purposes. Although the studies on internet-related expression of the condition did not regard animals specifically, the mode may similarly apply to situations when animals are proxy. Thus, further studies are needed to assess prevalence aspects of AAF and to validate which signs and symptoms in client behaviour and relating to animals and conditions are telling of AAF and which may be most suitable as warning signs. We note that the signs and symptoms listed in this study unlikely cover all possible warning signs of AAF. AAF may present itself highly varied in the veterinary clinical setting, as it does in the human medical setting [2].

We stress that further studying of AAF may have an additional possible importance. If future studies can contribute to creating AAF guidelines and support the veterinary profession in early recognition of possible AAF cases, this is of value to not only animal welfare, but may have broader societal implications. Cases are known in which both animals and children are a proxy [13,25]. Already in 1998, a case was described where in one household child and animal were proxy, in a study on male perpetrators of child abuse by falsification [25]. And, in 2001, an AAF case was described where dog poisoning surfaced after the poisoning of a child came to light [13]. Thus, studying AAF may have a broader value than animal welfare value. With the continuously growing body of literature on the relation between animal abuse and child abuse [26–28], the studying of AAF may contribute to (early) recognition of abuse cases and possibly more complex situations of intergenerational abuse effects [29,30].

## Conclusion

Animal abuse by (Condition) Falsification (AAF) is recognised by a part of the Dutch veterinary profession and warning signs of the phenomenon are known to some extent. Yet, presently little guidance exists for (early) recognition of these particular animal abuse cases. A support need could possibly be filled, at least in part, by creating guidelines on the assessment and management of AAF. Before such guidelines can be established, the further studying of AAF, its prevalence and (warning) signs is needed as an initial step to support the veterinary profession.

## Supporting information

**S1 File. Survey items on Animal Abuse by Falsification (AAF).**
(DOCX)

**S2 Table. Literature sources for Animal Abuse by Falsification (AAF) signs and symptoms.**
(DOCX)

**S3 Table. Participant details.**
(DOCX)

**S4 Table. Animal Abuse by Falsification (AAF) signs and symptoms mentioned by participants to the open question on recognition: 'Which signs would make you think of AAF?'.**
(DOCX)

**S5 Table. What would aid in reporting Animal Abuse by Falsification (AAF) as mentioned by participants in the open question on reporting: 'What would help you to report AAF?'.**
(DOCX)

**S1 Data. Data file.**
(XLSX)

## Author contributions

**Conceptualization:** Ineke R. van Herwijnen, Daphne G.L. van Helvoort.

**Formal analysis:** Ineke R. van Herwijnen.

**Investigation:** Daphne G.L. van Helvoort.

**Methodology:** Ineke R. van Herwijnen, Daphne G.L. van Helvoort.

**Writing – original draft:** Ineke R. van Herwijnen.

**Writing – review & editing:** Ineke R. van Herwijnen, Nadieh Reinders, Claudia M. Vinke.

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
