## [Decision Letter · Decision Letter 0]

1 Feb 2026

Dear Dr. Herwijnen,

Thank you for submitting your manuscript to PLOS ONE. After careful consideration, we feel that it has merit but does not fully meet PLOS ONE’s publication criteria as it currently stands. Therefore, we invite you to submit a revised version of the manuscript that addresses the points raised during the review process.

We look forward to receiving your revised manuscript.

Kind regards,

Selim Adewale Alarape, MVPH

Academic Editor

PLOS One

Journal Requirements:

2. Please include a new copy of Table S4 in your manuscript; the current table is difficult to read. Please follow the link for more information: https://journals.plos.org/plosone/s/tables

Reviewers' comments:

Reviewer's Responses to Questions

**Comments to the Author**

1. Is the manuscript technically sound, and do the data support the conclusions?

Reviewer #1: Yes

Reviewer #2: Yes

2. Has the statistical analysis been performed appropriately and rigorously?

Reviewer #1: Yes

Reviewer #2: Yes

3. Have the authors made all data underlying the findings in their manuscript fully available?

Reviewer #1: Yes

Reviewer #2: Yes

4. Is the manuscript presented in an intelligible fashion and written in standard English?

Reviewer #1: Yes

Reviewer #2: No

Reviewer #1: The context of the article titled "Animal abuse by falsification – recognition amongst the veterinary profession in The Netherland" is highly appreciable in this era of the sociological need to assess human-animal interactions. The novelty is awesome. However, there is need to look at the following areas for possible amendments to add value to the write up:

1. Line 33-35: There is need for modification and or recapping of the statement. The present forms is more of history than science!. What is the problem statement?. Highlight the gap in knowledge with this sentence.

2. Line 38: What is vet tech? Please, write fully for readers who are not from your field to understand.

3.Line 50-51: What is the essence of the last sentence?

4. Line 78: .......is unknown. Where? Mention the name of the country

5. Line 119: Participants

6. Line 123: What is vet tech??

7. The conclusion is not shown as a distinct subheadings like Results and Discussion. Please, carve the Conclusion out of the discussion.

8. No DOI for the articles listed under the references.

Reviewer #2: Line 36-" perplexing presentation has been mentioned'' include reference so we know who is being referred to.

Line 46- Reporting of AAF in this sample was low. this statement is too short and should be merged with previous statement

Line 47- does the 92% reported here inclusive of the participants that indicated they are unsure and have not seen such cases? I dont think these categories should be in this

line 59- recast the sentence like outcome may include instead of outcome may regard

Line 67: among not amongst

line 74- cant start a sentence with or, please recast or merge

line 77-82: recast the sentence to have a better flow,

Line 85- switching among

Line 178- majority were veterinarians delete was a

Lines 110 and 120- Repeating the same information. Can you please recast the sentence for better clarity to be like we recruited...................... via e mail and platform

Line 164- reframe the sentence ''how likely they thought a case to regard AAF,'

line 200- 31.8% cannot be termed more since 51.2 % already indicated, please remove more and recast the sentence

-- Create a footnote for table 1, what is PU-PD

290-291- e.g. (4) what is the meaning of this? please clarify

Line 291- This, as some participants responded that they recognise as an indication of AAF'' please recast for clarity

Line 297-298-- statement is disjointed, please recast

line 304- please recast

**Do you want your identity to be public for this peer review?** For information about this choice, including consent withdrawal, please see our Privacy Policy

Reviewer #1: **Yes:** Olufemi Mobolaji Alabi P.hD

Reviewer #2: **Yes:** Bukola O. Oyebanji

---

## [Author Response · Author response to Decision Letter 1]

10 Feb 2026

Response to each point raised:

2. Please include a new copy of Table S4 in your manuscript; the current table is difficult to read. Please follow the link for more information: https://journals.plos.org/plosone/s/tables

Thank you for suggesting this change, we have tried several variants. We have now added a supplement table with larger font size, while still keeping the data in one table. Keeping the information in one table was deemed beneficial by our test readers.

Reviewer #1: The context of the article titled "Animal abuse by falsification – recognition amongst the veterinary profession in The Netherland" is highly appreciable in this era of the sociological need to assess human-animal interactions. The novelty is awesome. However, there is need to look at the following areas for possible amendments to add value to the write up:

Thank you for this appreciated feedback, your reviewing time and suggestions, that helped us improve the manuscript.

1. Line 33-35: There is need for modification and or recapping of the statement. The present forms is more of history than science!. What is the problem statement?. Highlight the gap in knowledge with this sentence.

We have adapted our presentation of the topic at hand in the first three lines of the abstract.

2. Line 38: What is vet tech? Please, write fully for readers who are not from your field to understand.

Thank you for this suggestion, we have replaced ‘tech’ with ‘technician’ throughout.

3.Line 50-51: What is the essence of the last sentence?

We have removed the last sentence of the abstract.

4. Line 78: .......is unknown. Where? Mention the name of the country

We have added the USA region.

5. Line 119: Participants

We have adapted the title of this section.

6. Line 123: What is vet tech??

Thank you for this suggestion, we have replaced ‘tech’ with ‘technician’ throughout.

7. The conclusion is not shown as a distinct subheadings like Results and Discussion. Please, carve the Conclusion out of the discussion.

We have added a conclusion section.

8. No DOI for the articles listed under the references.

We have added the DOI to the articles in the reference list.

Reviewer #2:

Thank you for your time and comments that helped to improve our manuscript.

Line 36-" perplexing presentation has been mentioned'' include reference so we know who is being referred to.

Thank you for your suggestion. We are not customary to adding references to an abstract, but have provided references in the Introduction section.

Line 46- Reporting of AAF in this sample was low. this statement is too short and should be merged with previous statement

We have merged this sentence and the sentence thereafter.

Line 47- does the 92% reported here inclusive of the participants that indicated they are unsure and have not seen such cases? I dont think these categories should be in this

Reporting was indeed reportedly never done by 92.1% (N=81), 4.5% (N=4) reported AAF as animal abuse, 3.4% (N=3) didn’t know.

line 59- recast the sentence like outcome may include instead of outcome may regard

Thank you for this suggestion, which we adhered to.

Line 67: among not amongst

As far as we are aware, amongst is more commonly used in British English, but we have adapted to among, hoping this is in accord with PLOS One.

line 74- cant start a sentence with or, please recast or merge

Thank you for this suggestion, which we adhered to.

line 77-82: recast the sentence to have a better flow,

We have rewritten this section. It now reads: ‘A USA-based study indicated that the true incidence or prevalence of AAF is unknown [2]. The few available scientific studies on the topic, often present cases and do not provide epidemiological data [2]. An example is a UK-based study including six cases identified by veterinarians and three cases included by the researchers as possible AAF cases [13]. In this total of nine cases, signs were noted in the animals, that corresponded with signs in child abuse cases. The respective signs regarded the animal owner’s attention seeking behaviour, recovery of the animal upon owner-separation, serial incidents, real and likely factitious clinical signs, abnormal biochemical profiles, deliberately inflicted injury, tempering with surgical sites and switching from veterinary clinics [13].’

Line 85- switching among

This sentence was adapted.

Line 178- majority were veterinarians delete was a

This sentence was adapted.

Lines 110 and 120- Repeating the same information. Can you please recast the sentence for better clarity to be like we recruited...................... via e mail and platform

Unfortunately PLOS One requires the ethical statement to be formulated as is and our reading panel judged it necessary to present the information that you point at also at the second location to facilitate the readers.

Line 164- reframe the sentence ''how likely they thought a case to regard AAF,'

This sentence was adapted.

line 200- 31.8% cannot be termed more since 51.2 % already indicated, please remove more and recast the sentence

Thank you for pointing out the unclarity in our writing. We have rewritten this text as: ‘51.2% (N=45) indicated to likely see cases. 4.6% (N=4) indicated to certainly see cases, 12.5% (N=11) indicated to certainly not see cases in their clinic. A relatively high percentage of 31.8% (N=28) indicated ‘don’t know’.’

-- Create a footnote for table 1, what is PU-PD

Thank you for this suggestion. We added ‘polyuria-polydipsia’ between brackets behind the abbreviation.

290-291- e.g. (4) what is the meaning of this? please clarify

We have rewritten this text section, taking away the unclarity.

Line 291- This, as some participants responded that they recognise as an indication of AAF'' please recast for clarity

We have rewritten this text section, taking away the unclarity.

Line 297-298-- statement is disjointed, please recast

We have rewritten this text section, taking away the unclarity.

line 304- please recast

This text was adapted.

---

## [Editor Report · Decision Letter 1]

2 Mar 2026

Animal abuse by falsification – recognition amongst the veterinary profession in the Netherlands

PONE-D-25-67684R1

Dear Dr. Herwijnen,

We’re pleased to inform you that your manuscript has been judged scientifically suitable for publication and will be formally accepted for publication once it meets all outstanding technical requirements.

Kind regards,

Selim Adewale Alarape, MVPH

Academic Editor

PLOS One
---

## [Editor Report · Acceptance letter]

PONE-D-25-67684R1

PLOS One

Dear Dr. Herwijnen,

I'm pleased to inform you that your manuscript has been deemed suitable for publication in PLOS One. Congratulations! Your manuscript is now being handed over to our production team.

Kind regards,

on behalf of

Dr. Selim Adewale Alarape

Academic Editor

PLOS One